# Salivary BCAA, Glutamate, Glutamine and Urea as Potential Indicators of Nitrogen Metabolism Imbalance in Breast Cancer

**DOI:** 10.3390/cimb47100837

**Published:** 2025-10-11

**Authors:** Elena A. Sarf, Lyudmila V. Bel’skaya

**Affiliations:** Biochemistry Research Laboratory, Omsk State Pedagogical University, 644099 Omsk, Russia; sarf_ea@omgpu.ru

**Keywords:** saliva, breast cancer, BCAA, glutamate, glutamine, urea, nitrogen metabolism

## Abstract

Nitrogen metabolism in the human body is in a strictly balanced state, which is disturbed in pathologies, including breast cancer. The state of nitrogen balance can be judged by the content of urea and the amount of branched-chain amino acids (BCAAs) (Val, Leu, and Ile), glutamine (Gln), and glutamate (Glu). The study involved 1438 people, including patients with breast cancer (n = 543), fibroadenomas (n = 597), and healthy controls (n = 298). Saliva samples were collected from all patients before treatment, and urea levels were determined in all 1438 samples. Salivary levels of BCAAs, Gln, and Glu were determined in 116 patients with breast cancer, 24 with fibroadenomas, and 25 healthy volunteers. An increase in the concentration of urea in saliva was shown in breast cancer, most pronounced in luminal molecular biological subtypes: luminal A 10.46 [7.69; 12.62] mmol/L (*p* < 0.0001), luminal B HER2-negative 9.52 [6.72; 12.52] mmol/L (*p* = 0.0198), and luminal B HER2-positive 8.26 [5.27; 12.07] mmol/L. The Gln/Glu ratio increased in the saliva of the control group (5.43 [3.30; 10.5]) compared with breast cancer (2.22 [0.84; 5.40], *p* = 0.0094) and fibroadenomas (1.94 [0.89; 6.05], *p* = 0.0184). For luminal B HER2-positive and TNBC, the Gln/Glu ratio increased sharply to 8.23 [3.24; 10.9] (*p* = 0.0327) and 11.2 [4.28; 15.2] (*p* < 0.0001) compared with healthy controls. Thus, an increased Gln/Glu ratio in saliva may characterize a more aggressive subtype of breast cancer.

## 1. Introduction

The content of free nitrogen depends on the balanced catabolic and anabolic processes occurring in the body [1]. The urea cycle is the major pathway for nitrogen metabolism [2]. In destructive processes, protein breakdown increases the level of free nitrogen, which is converted into urea and excreted from the body with urine or through the salivary glands [3,4]. Measuring urea levels in saliva is actively used to assess kidney health, but can also be informative for a number of other pathologies [5]. Thus, in cancer, inactivation of specific enzymes in the urea cycle can lead to increased nitrogen utilization for pyrimidine synthesis and maintenance of RNA and DNA synthesis in tumor cells, which accelerates tumor development [6]. Thus, in our opinion, studying the relationship between nitrogen utilization and cancer development is promising [7].

Another form of reflecting the amount of nitrogen is measuring the content of glutamine (Gln) and glutamate (Glu) in saliva. These amino acids are involved in the binding, accumulation, and transportation of nitrogen (Gln) and reflect the process of releasing nitrogen for the synthesis of replaceable amino acids de novo (Glu). Gln can release amide nitrogen to synthesize ammonia and convert it to Glu [8]. In addition, it can produce α-ketoglutarate and remove accumulated ammonia through deamination or transamination [9]. Branched-chain amino acids (BCAAs) such as valine (Val), leucine (Leu), and isoleucine (Ile) serve as a source for replenishing Gln [10,11]. Tumor cells require a large amount of nutrients and structural elements to ensure active proliferation, growth, and metabolism [12]. Increased plasma BCAA concentrations have been noted in breast cancer [13]. In addition to being essential amino acids, BCAAs are a source of nitrogen and carbon, supporting amino acid and nucleotide synthesis and energy balance [14]. How BCAA signaling molecules may influence tumor development by affecting epigenetics, oxidative stress, drug resistance, and immune responses [15,16].

During malignant transformation, including breast cancer, changes in amino acid and urea levels are observed both in tumor tissue and in the blood serum. Urea and amino acids can be transported across cell membranes via active transport and facilitated diffusion, allowing them to cross the blood–salivary barrier and enter saliva. Saliva composition has previously been shown to reflect systemic metabolic changes in breast cancer, including those taking into account the tumor’s molecular biological subtype [17]. Changes in saliva composition can be used to track changes in key components of nitrogen metabolism, which can be targeted for therapeutic intervention within metabolic therapy. The advantages of saliva as a biomaterial include not only cost benefits, ease of collection, transportation, and storage, but also the potential for the development of personalized medicine [18].

We hypothesize that the assessment of urea and amino acid levels in saliva may provide important information about the characteristics of nitrogen metabolism in breast cancer, including different phenotypes, which may be useful for both diagnostic and therapeutic purposes. The aim of the study is to evaluate the change in the levels of urea, BCAAs, Gln, and Glu in saliva in patients with breast cancer.

## 2. Materials and Methods

### 2.1. Study Design

A total of 1438 people were included in the case–control study, including the following subgroups: breast cancer (BC, n = 543), fibroadenomas (FA, n = 597), and healthy controls (HC, n = 298). The distribution of breast cancer phenotypes was as follows: triple negative breast cancer (TNBC)—97 (17.8%), luminal A—123 (22.7%), luminal B (HER2-negative)—174 (32.0%), luminal B (HER2-positive)—47 (8.7%), and non-luminal (HER2-negative)—47 (8.7%). In healthy volunteers, no breast pathologies were detected during routine mammographic and ultrasound examinations. All patients had their saliva urea concentration determined.

Determination of amino acid concentrations was performed in 165 patients: BC (n = 116, age 56.6 ± 2.3 years), FA (n = 24, age 47.0 ± 4.5 years), and HC (n = 25, age 49.7 ± 4.1 years). Breast cancer patients had the following phenotypes: TNBC—14 (12.1%), luminal A—40 (34.5%), luminal B (HER2-negative)—35 (30.2%), luminal B (HER2-positive)—15 (12.9%), and non-luminal (HER2-negative)—12 (10.3%).

Participants were enrolled in the groups simultaneously. Inclusion criteria included female gender, age 30–70 years, the absence of any treatment at the time of the study, including surgery, chemotherapy, or radiation, no signs of active infection (including purulent processes), and oral hygiene. Exclusion criteria included the absence of histological verification of the diagnosis. A physician excluded chronic, inflammatory, and infectious diseases during a routine medical examination. A dental examination was also conducted to rule out any inflammatory oral diseases that could affect the saliva analysis results.

### 2.2. Collection and Storage of Saliva Samples

Saliva samples were obtained before the start of surgical or chemotherapy treatment. Sterile 10-mL polypropylene centrifuge tubes with screw caps were used for sample collection. Unstimulated saliva samples were collected in the morning, on an empty stomach, after rinsing the mouth with distilled water. Immediately after collection, samples were centrifuged at 10,000× *g* for 10 min (CLb-16, Moscow, Russia). Urea concentration in saliva was determined without storage or freezing for 4–6 h after collection. A 1-mL aliquot was transferred to Eppendorf tubes after centrifugation and stored in a freezer at −80 °C until amino acid analysis.

### 2.3. Determination of the Composition of Saliva

The concentration of urea in saliva was determined by an enzymatic method using the commercial Vector-Best kit (cat.No. B-8074, Novosibirsk, Russia). Urea is decomposed by urease into carbon dioxide and ammonia. Ammonia reacts with sodium salicylate and sodium hypochlorite in the presence of sodium nitroprusside to form a colored product. The color intensity of the reaction product at a wavelength of 578 nm is proportional to the concentration of urea in the sample (mmol/L). A calibrator with a concentration of 8.33 mmol/L was used to calculate the urea concentration; the coefficient of variation was no more than 5%. The accuracy of the determination was verified using control sera from Vector-Best LLC (Novosibirsk, Russia), certified for this method. All analyses were performed in duplicate.

The amino acid composition of saliva was analyzed using high-performance liquid chromatography in the selected reaction monitoring mode on a 1260 Infinity II chromatograph with detection on a 6460 Triple Quad mass spectrometer (Agilent, Santa Clara, CA, USA). An HPLC method with mass spectrometric detection in selected reaction monitoring mode was developed for the analysis of test compounds in samples. An internal standard (alanine-d4) was used for the back-calculation of concentrations. Calibration curves were linear functions with a correlation coefficient of *R*^2^ ≥ 0.98. At least six samples of the Amino Acid Kit (Jasem, Istanbul, Turkey) were used to construct the calibration scale. The accuracy of the calibration curve response was verified by back-calculating the concentrations of the calibration standards. The variation coefficients were 2.78% for Leu + Ile, 7.12% for Glu, 4.58% for Gln, and 6.36% for Val.

Saliva samples were collected for 10 min, after which the volume was recorded and the salivary flow rate (mL/min) was calculated. Total protein concentration in saliva (g/L) was also determined by the pyrogallol red reaction using a commercial Vector-Best kit (Novosibirsk, Russia). No significant differences were shown between the subgroups, so the data are not discussed in the text of the manuscript.

### 2.4. Statistical Analysis

Statistical analysis was performed using Statistica 13.3 EN software (StatSoft, Tulsa, OK, USA) by a nonparametric method. The results were presented as the median and the interquartile range. The Bonferroni correction was applied as a *p*-value adjustment when comparing more than two subgroups: instead of adjusting the alpha level, each *p*-value was multiplied by the number of tests, and the alpha level was left unchanged (*p* < 0.05).

## 3. Results

The concentration of salivary urea in the BC group was significantly higher compared to the group of patients with FA and HC (Table 1). One of the factors determining the differences in the concentration of urea in the compared groups can be considered the difference in age (Table 1). We calculated the correlation of the concentration of salivary urea with age using the Spearman method and showed the presence of a weak correlation with age. In HC, the correlation was *r* = 0.3116 (*p* < 0.0001); in FA, *r* = 0.2325 (*p* < 0.0001); and in BC, *r* = 0.2182 (*p* < 0.0001). Additionally, each group was divided into subgroups, taking into account the status of menopause. It was found that the salivary urea concentration in HC depended weakly on menopause (6.40 [4.57; 8.95] and 6.76 [4.84; 9.29] mmol/L before and after menopause, respectively). For the FA group, an increase in the concentration of urea in saliva was observed after menopause (7.39 [5.00; 10.37] and 9.39 [6.84; 12.68] mmol/L before and after menopause, respectively). Similarly, for the BC group, an increase in the concentration of urea in saliva was shown after menopause (8.22 [5.70; 11.96] and 10.30 [7.07; 13.94] mmol/L before and after menopause, respectively). The differences from HC were statistically significant in all cases: before menopause, FA vs. HC (*p* = 0.0029) and BC vs. HC (*p* = 0.0001); after menopause, FA vs. HC (*p* < 0.0001) and BC vs. HC (*p* < 0.0001). Thus, with a correct comparison of subgroups taking into account menopause, a statistically significant increase in the concentration of urea in saliva in the series “HC–FA–BC” was preserved.

No correlation with age and menopause status was shown for the concentration of free amino acids in saliva. Thus, for Leu + Ile, the correlation coefficients with age were *r* = 0.0552, *r* = 0.1186, and *r* = −0.0123 for BC, FA, and HC, respectively (*p* ˃ 0.05). The correlation coefficients were for Val *r* = 0.1780, *r* = −0.0141, and *r* = −0.0649; for Glu *r* = 0.1767, *r* = 0.0243, and *r* = 0.1615; for Gln *r* = −0.1034, *r* = 0.1780, and *r* = 0.1464 in the corresponding three subgroups (*p* ˃ 0.05). Division into subgroups by menopause status also did not show significant differences in the content of free amino acids in saliva, so further calculations were carried out without additional division into subgroups.

In patients with BC and FA, the concentrations of amino acids in saliva were higher than in the HC group, with the exception of Gln (Table 1). Thus, in FA, the increase in concentration was for Glu +25.9% and Val +21.4%, whereas in BC, the increase in concentration was observed for Glu +33.9%, Val +27.3% and Leu + Ile +112.4%. Only for Gln was a decrease in concentration in saliva observed in BC and FA compared to the control group (−45.6% and −58.8%, respectively). Moreover, the content of Glu and Gln in saliva in BC was higher than in FA (+4.6% and +32.2%, respectively), whereas the content of Val, Leu + Ile in saliva was lower than in FA (−4.9% and −221.2%, respectively) (Table 1).

Next, the correlation relationship was calculated using the Spearman method between the content of BCAAs (Val, Leu + Ile), Glu, Gln, and urea in BC. A statistically significant direct correlation was shown between Glu and urea (*r* = 0.2300, *p* = 0.0170), as well as between Val and urea (*r* = 0.4100, *p* < 0.001) in BC. Leu + Ile and Gln did not show correlation dependence with urea (Figure 1).

When calculating the correlation dependence between amino acids and urea depending on the molecular biological subtype of BC, a direct statistically significant correlation was shown only in the group with the luminal A subtype between Glu and urea (*r* = 0.3971, *p* = 0.0306). In FA, a statistically significant direct correlation was shown between Gln and urea (*r* = 0.6485, *p* = 0.0490). In HC, no statistically significant correlation was shown between the BCAAs (Val, Leu + Ile), Glu, Gln, and urea.

The highest concentration of salivary urea was observed in luminal BC subtypes: luminal A—10.46 [7.69; 12.62] mmol/L (*p* < 0.0001), luminal B HER2-negative—9.52 [6.72; 12.52] mmol/L (*p* = 0.0198), and luminal B HER2-positive—8.26 [5.27; 12.07] mmol/L. For non-luminal HER2-negative and TNBC, the concentration of urea in saliva did not differ from the HC (7.20 [4.72; 9.86] and 6.87 [4.08; 10.48] mmol/L, respectively) (Figure 2).

Next, we analyzed how the amount of salivary amino acids changes depending on the BC subtype relative to each other and the control values. The concentration of Glu in saliva statistically significantly increased in luminal A (+45.3%, *p* = 0.0204) and luminal B HER2-positive (+33.6%, *p* = 0.0212) compared to the HC. In contrast, the concentration of Gln was maximally reduced in the saliva of patients with luminal A subtype compared to the HC (−70.9%, *p* = 0.0447) and other molecular biological subtypes of BC (Figure 3). In patients with luminal B HER2-positive and TNBC, the concentration of Gln in saliva was increased compared to other molecular biological subtypes and HC (+50.7% and +45.4%). The BCAA (Val, Leu + Ile) content in saliva was significantly higher in luminal A subtype compared to other BC subtypes (Figure 3C,D). For luminal B HER2-positive (3.16 times, *p* = 0.0036) and TNBC (2.58 times, *p* = 0.0042), a statistically significant increase in the concentration of Ile + Leu in saliva was shown compared to FA (Figure 3D).

Additionally, the Gln/Glu ratio was calculated. It was shown that the Gln/Glu ratio was higher in the saliva of the HC (5.43 [3.30; 10.5]) compared to BC (2.22 [0.84; 5.40], *p* = 0.0094) and fibroadenomas (1.94 [0.89; 6.05], *p* = 0.0184). The Gln/Glu ratio differed for BC phenotypes. Thus, the minimum ratio was characteristic of luminal A (1.26 [0.84; 3.63], *p* = 0.0057), luminal B HER2-negative (2.61 [0.77; 5.27]), and non-luminal BC (3.23 [1.45; 5.40]). For luminal B HER2-positive and TNBC, the Gln/Glu ratio increased sharply to 8.23 [3.24; 10.9] (*p* = 0.0327) and 11.2 [4.28; 15.2] (*p* < 0.0001) compared with HC.

## 4. Discussion

Urea is secreted into the oral cavity by the minor salivary glands and, to a lesser extent, the parotid and submandibular glands. Urea concentrations were higher in parotid saliva than in whole saliva or plasma. It can be hypothesized that the mechanism of urea entry into parotid saliva is active transport, involving urea transporters (UT-A and UT-B), although their presence in the salivary glands has not yet been confirmed [19]. Urea is the most abundant (non-protein) nutrient in saliva, utilized by bacteria such as *Streptococcus salivarius, Actinomyces naeslundii*, and *haemophilus*, presumably through the expression of urease, an enzyme that converts urea to ammonia and carbon dioxide [20]. However, another study using C13-labeled urea found that urea was first converted to ammonium carbamate and then to formate and propionate, suggesting that some bacteria convert urea to formate and some to ammonia [20].

A positive correlation between urea levels in blood and saliva has been shown in a number of studies in kidney disease [21]. The concentration of urea in saliva was associated with the severity of kidney disease [22] and decreased with hemodialysis [23]. This suggests that the concentration of salivary urea adequately reflects its content in the blood and can be used for diagnosis.

We showed that the concentration of salivary urea increased in breast diseases, and in breast cancer, the urea content in saliva reached its maximum values. The concept of multiple organ microbiota axes, including the oral-gut-liver microbiome axis, has been widely publicized [24,25]. The relationship between the oral and intestinal microbiomes indicates a dynamic interaction of microbial communities, which in particular influences the development and progression of cancer [26,27,28,29,30]. Thus, alpha diversity and the presence/relative abundance of certain genera in the oral microbiome were strongly associated with breast cancer [31]. It can be assumed that an increase in the concentration of urea in saliva in breast cancer is due to a decrease in the diversity of the oral microbiome. It should be noted that the highest concentration of urea in saliva was observed in luminal subtypes of breast cancer, which may be due to the presence of expression of estrogen receptors for these subtypes, which in turn is associated with changes in the composition of the microbiome. Many bacteria can encode genes for enzymes that metabolize estrogen, thereby affecting the regulation of estrogen levels in the blood serum [32]. On the other hand, estrogen-like compounds can stimulate the growth and development of some bacteria. It has also been shown that the oncobiome of each subtype of breast cancer is unique and contains a wide range of microbial features [33]. The most diverse oncobiome was in estrogen receptor-positive cancer, and the least diverse in triple-negative cancer [34].

Higher serum urea concentrations have been shown to protect against overall cancer incidence and cancer mortality [35], which is consistent with the finding that salivary urea concentrations are elevated in the most prognostically favorable luminal A and B HER2-negative breast cancers.

Nitrogen metabolism is characterized not only by the rate of urea formation, but also by a number of amino acids. Their metabolism occurs in two stages: transamination and then oxidative deamination with the formation of the final product ammonia [36]. Any amino acid and keto acid participate in transamination with the formation of a new amino acid and Glu [37]. Then, Glu enters into an oxidative deamination reaction with the formation of free ammonia and alpha-keto acid. Ammonia interacts with carbon dioxide, forming carbamoyl phosphate for the synthesis of urea, and alpha-keto acid can decompose to water and carbon dioxide, leading to the synthesis of glucose, fatty acids, or ketone bodies. The Glu level reflects the rate of amino acid synthesis de novo [38]. It is known that amino acids are not capable of accumulating, but a constant and sufficient amount of nitrogen is necessary to maintain catabolic processes. Gln acts as an amino acid capable of accumulating nitrogen. Since free ammonia is toxic to the body, it binds to Glu under the action of glutamine synthetase to form Gln, and then enters the bloodstream into the liver, where it breaks down into free ammonia and Glu under the action of glutaminase. After this, Glu in the bloodstream either re-interacts with ammonia or participates in the transamination reaction for the synthesis of amino acids de novo [39]. It is important to note that branched-chain amino acids (BCAAs) are not metabolized in the liver and are able to replenish the Gln pool [40].

In our study, the level of free amino acids increased in saliva in breast cancer, with the exception of Gln. It is known that there is an increase in the content of amino acids in breast cancer tissues, which in turn leads to an increase in the content of amino acids in the blood [41]. Amino acids can be transported through cell membranes by active transport and facilitated diffusion, which causes them to pass through the hematosalivary barrier and enter the saliva.

In patients with luminal A and luminal B HER2-negative subtypes of breast cancer, we observe a high content of urea in saliva with a simultaneously high content of Glu and BCAAs; the concentration of Gln was lower than the control values. This indicates both active protein breakdown because of the oncological process and the breakdown of short-lived proteins of the immune defense. Low levels of Gln in saliva may indicate both its accumulation in the liver to release free ammonia for urea synthesis and its direct absorption by cells to meet increased metabolic needs. At the same time, the amount of free BCAAs was high compared to the control group. This indicates that the Gln pool was not replenished by BCAA concentrations.

The highest BCAA concentrations in saliva were observed in luminal B HER2-positive and TNBC. BCAAs activate protein synthesis and promote breast cancer progression, which is observed for subtypes with high proliferative activity of breast cancer cells [42]. An increase in the concentration of Gln in saliva was noted for luminal B HER2-positive and TNBC, while no clear trend was observed for Glu. However, calculation of the ratio of Gln/Glu concentrations showed that, for luminal A and B HER2-negative subtypes, this ratio was less than 3, while for luminal B HER2-positive and TNBC, it exceeded 8. Thus, a significant predominance of Gln compared to Glu in saliva was observed in these subtypes of breast cancer. There are two main metabolic phenotypes of breast cancer: the first depends mainly on glycolysis and the pentose phosphate pathway and is associated with low survival (most typical for HER2-positive and TNBC), the second—on fatty acid oxidation and glutaminolysis (typical for the luminal A subtype of breast cancer) [43]. It is known that an increase in the level of amino acids correlates with pro-inflammatory and immunological factors and a more aggressive subtype of breast cancer [41].

In the group of patients with luminal B HER2-positive and TNBC, the urea level was reduced with a significant increase in Gln and free BCAAs in saliva. These subtypes of breast cancer are characterized by high proliferative activity, which indicates both damage to nearby tissues by a conglomerate of tumor cells with subsequent destruction of protein and high metabolic activity. In this case, we see an active redistribution of nitrogen as a valuable source of energy and building material that is not excreted from the body through the synthesis of urea.

Study limitations included the lack of information on smoking status, diet, and fluid intake, and medication use, which could have influenced saliva composition. We also did not determine the oral microbiota composition in this study, so the conclusions regarding the relationship between salivary urea concentration and the oral microbiome should be considered a hypothesis that requires further verification.

## 5. Conclusions

An increase in the concentration of urea in saliva was observed in breast cancer, most pronounced in luminal molecular biological subtypes. This may be the result of a change in the composition of the oral microbiome in breast cancer towards a decrease in diversity, and, accordingly, a decrease in the utilization of urea by oral bacteria. This process is estrogen-dependent, which is reflected in a more active change in the concentration of urea in saliva in luminal subtypes of breast cancer. An additional analysis of nitrogen metabolism in saliva was carried out using the BCAAs, Glu, and Gln as an example. In patients with luminal B HER2-positive and TNBC, the metabolism and redistribution of amino acids increased, which characterizes a more aggressive subtype of breast cancer. BCAAs replenish the Gln pool, which is one of the main sources of energy and building material for cancer cells. Thus, to assess the activity of catabolic processes in saliva in patients with breast cancer, an assessment of the amino acid balance may be more informative, whereas to assess changes in the oral microbiome, the concentration of urea in saliva can be used.

## Figures and Tables

**Figure 1 cimb-47-00837-f001:**
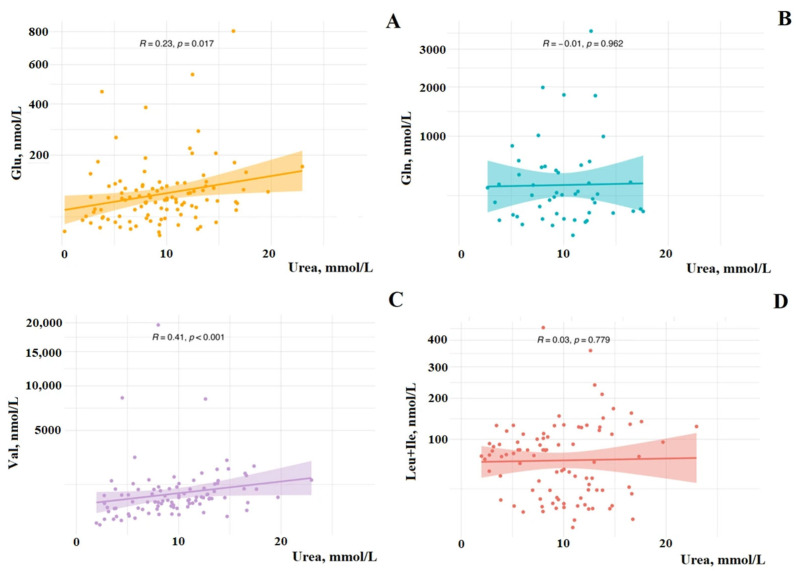
Spearman correlation between urea and amino acid content in saliva in BC: (**A**) glutamic acid (Glu), (**B**) glutamine (Gln), (**C**) valine (Val), (**D**) leucine (Leu) + isoleucine (Ile). R—Spearman correlation coefficient.

**Figure 2 cimb-47-00837-f002:**
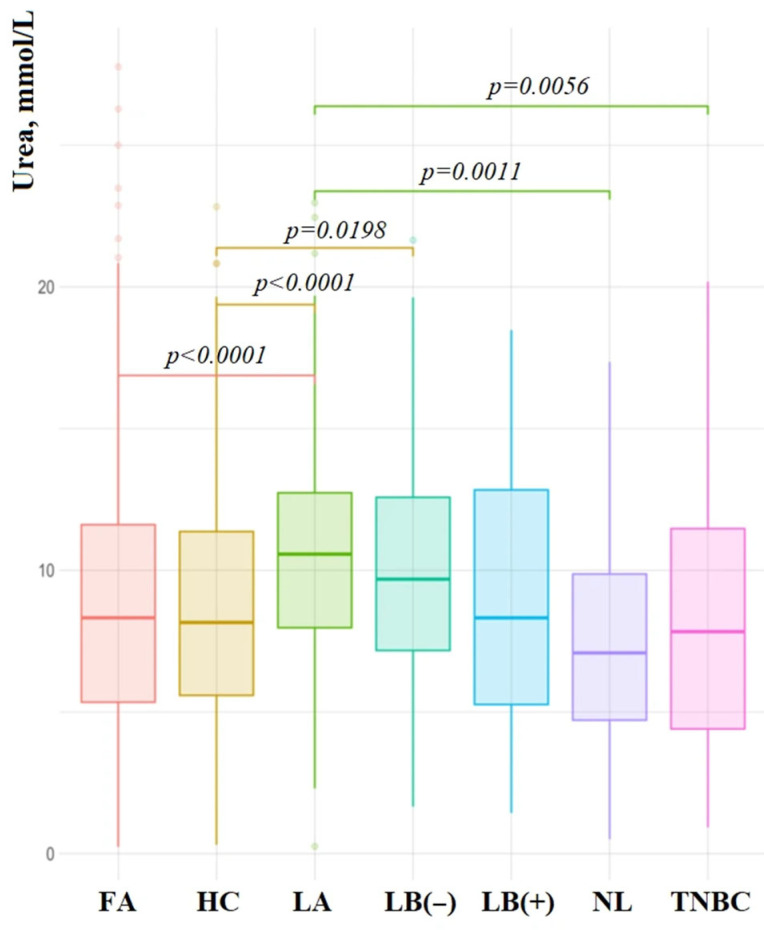
Urea content in saliva in different molecular biological subtypes of BC, FA, and HC. FA—fibroadenoma, HC—healthy control, LA—luminal A, LB(−)—luminal B HER2-negative, LB(+)—luminal B HER2-positive, NL—non-luminal, TNBC—triple negative breast cancer.

**Figure 3 cimb-47-00837-f003:**
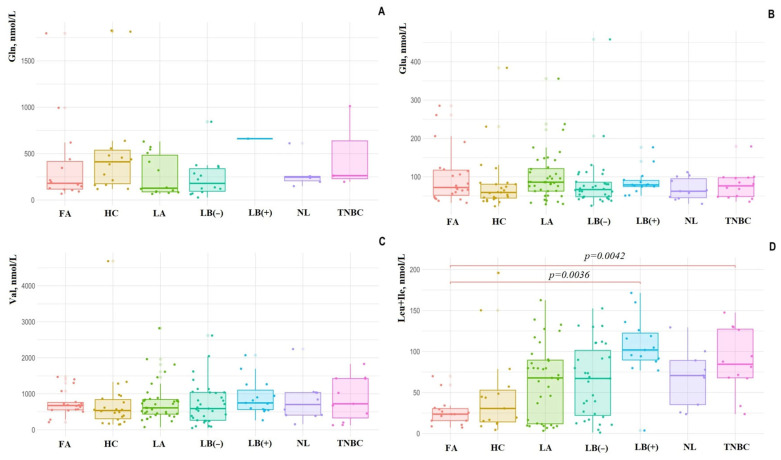
Amino acid concentrations in saliva in healthy controls, fibroadenomas, and breast cancer depending on phenotypes: (**A**) Gln; (**B**) Glu; (**C**) Val; (**D**) Leu + Ile.

**Table 1 cimb-47-00837-t001:** Concentrations of BCAAs (Val, Ile + Leu), Glu, Gln, and urea in saliva.

Indicator	BC, n = 543/116 * (1)	FA, n = 597/24 (2)	HC, n = 298/25 (3)
**Age, years**	54.7 ± 12.3	48.8 ± 12.8	46.3 ± 13.5
	*p*_1–2_ < 0.0001 *p*_1–3_ < 0.0001	-	-
**Urea, mmol/L**	9.77 [6.44; 13.31]	8.11 [5.15; 11.37]	6.67 [4.36; 9.13]
	*p*_1–3_ < 0.0001 *p*_1–2_ = 0.0001	*p*_2–3_ < 0.0001 *p*_2–3_ = 0.0001	-
**Glu, nmol/L**	77.9 [50.9; 102.9]	74.5 [51.8; 121.3]	59.2 [44.5; 80.6]
	*p*_1–3_ = 0.0424	-	-
**Gln, nmol/L**	238.8 [104.8; 412.8]	180.6 [114.5; 439.6]	438.8 [163.7; 638.4]
	*p*_1–3_ = 0.0050	-	-
**Val, nmol/L**	709.0 [408.9; 1041.0]	676.2 [551.3; 774.2]	557.1 [289.6; 944.9]
**Leu + Ile, nmol/L**	79.02 [34.42; 110.7]	24.50 [15.76; 32.77]	37.21 [14.17; 68.03]
	*p*_1–3_ = 0.0397 *p*_1–2_ = 0.0016	*p*_1–2_ = 0.0016	-

**Note.** Indices in *p*-values correspond to subgroups: «1»—breast cancer (BC), «2»—fibroadenomas (FA), «3»—healthy control (HC). *—The number of patients for whom urea/amino acid concentrations were determined.

## Data Availability

The data presented in this study are available on request from the corresponding author. The data are not publicly available due to privacy.

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
