# Peer review of "Salivary BCAA, Glutamate, Glutamine and Urea as Potential Indicators of Nitrogen Metabolism Imbalance in Breast Cancer"

_cimb, 2025, doi:10.3390/cimb47100837_

Round 1
Reviewer 1 Report
Comments and Suggestions for Authors
After thorough reading of the manuscript regarding nitrogen metabolism, and detection of related biomarkers in breast cancer patients, below you can find my suggestions:
Important Note: There are entire paragraphs in the article and especially at Materials and Methods’ and Results’ section detected as plagiarism. This issue needs to urgently be arranged.
Abstract
Please add the full name of BCAA
Line 12, add the term subjects
Based on the fact that the levels of aa were not determined at all subjects, the relevant phrase (line 13) should be changed
The symbol’’-‘’ before urea levels needs to be removed
Lines 18-20 representing a discussion section should be removed from the results, as it is rather confusing here
Again, lines 20-25 need modification to express the major findings of the present work and relate it with pre-existing knowledge
Introduction
The introductory part is not informative enough not at the biochemical and molecular mechanisms of urea and relevant amino acids but also, and more importantly at the relevant knowledge on nitrogen metabolism and cancer. Reference No.8 that is used is not related at the topic. These parts need enhancement (a figure also is proposed) in order the novelty of the present clinical study to be stated clearly.
Results
P1-2 and p1-3, at table 1 should be explained
The results should be analyzed in more detail, explaining each scheme, comparison and rational behind
Discussion
The assumptions made between the clinical results of this study and microbiome data of other studies should be analyzed in detail and enhanced with more bibliographic data
Author Response
Responses to reviewer’s comments_1
The authors are grateful to the reviewers for careful consideration of the manuscript and valuable comments. We hope that thanks to the joint work, the manuscript has become better.
Important Note: There are entire paragraphs in the article and especially at Materials and Methods’ and Results’ section detected as plagiarism. This issue needs to urgently be arranged.
We have made changes that should reduce the number of borrowings in the manuscript.
Abstract
Please add the full name of BCAA
We have made appropriate changes to the text of the manuscript.
Line 12, add the term subjects
We have made appropriate changes to the text of the manuscript.
Based on the fact that the levels of aa were not determined at all subjects, the relevant phrase (line 13) should be changed
We have made appropriate changes to the text of the manuscript.
«The study involved 1438 people, including patients with breast cancer (n=543), fibroadenomas (n=597) and healthy controls (n=298). Saliva samples were collected from all patients before treatment, and urea levels were determined in all 1,438 samples. Salivary levels of BCAA, Gln, and Glu were determined in 116 patients with breast cancer, 24 with fibroadenomas, and 25 healthy volunteers».
The symbol’’-‘’ before urea levels needs to be removed
We have made appropriate changes to the text of the manuscript.
Lines 18-20 representing a discussion section should be removed from the results, as it is rather confusing here.
We have removed lines 18-20 from the abstract.
Again, lines 20-25 need modification to express the major findings of the present work and relate it with pre-existing knowledge
We have made appropriate changes to the text of the manuscript.
Introduction
The introductory part is not informative enough not at the biochemical and molecular mechanisms of urea and relevant amino acids but also, and more importantly at the relevant knowledge on nitrogen metabolism and cancer. Reference No.8 that is used is not related at the topic. These parts need enhancement (a figure also is proposed) in order the novelty of the present clinical study to be stated clearly.
We have made appropriate changes to the text of the manuscript.
Results
P1-2 and p1-3, at table 1 should be explained
We have added a Note after the table with relevant explanations.
The results should be analyzed in more detail, explaining each scheme, comparison and rational behind
We have added more detailed explanations in the Results section.
Discussion
The assumptions made between the clinical results of this study and microbiome data of other studies should be analyzed in detail and enhanced with more bibliographic data.
We have provided numerous bibliographic references to recent studies on the salivary microbiome [24-34]. Since we did not directly study the oral microbiome, our conclusions can be considered purely hypotheses that require confirmation. We have included this information in the study's limitations and will continue to work in this direction.

Reviewer 2 Report
Comments and Suggestions for Authors
The manuscript examines salivary urea and amino acids (BCAA, glutamate, and glutamine) as non-invasive biomarkers for breast cancer and its molecular subtypes. The concept is clinically relevant. However, methodological inconsistencies, overinterpretation of results, and missing details limit the strength of the conclusions. Substantial revisions are required to improve clarity, rigor, and reproducibility.
- The abstract and methods state BC n=543, FA n=597, HC n=298, with amino acids in a subset (BC n=116, FA n=24, HC n=25). Yet, Table 1 lists BC (n = 660), FA (n = 134), and HC (n = 127). These discrepancies must be resolved, and the sample sizes for each analyte must be clearly stated.
- BC patients are older on average; urea correlates with age and menopause. While stratification by menopause is shown for urea, the amino acid results are not adjusted. Regression models that adjust for age, menopause, and other covariates (such as renal disease, oral health, diet, smoking, and hydration) are needed. Otherwise, findings may reflect age-related or renal differences rather than cancer.
- Urea and amino acids are influenced by kidney function, diet, and oral conditions. The manuscript lacks data on renal status, oral disease, medications, or dietary intake. These should be reported or at least discussed as limitations.
- The LC-MS/MS and enzymatic assays lack critical QC information: use of isotope-labeled standards, calibration ranges, intra-/inter-assay CVs, replicate analyses, and handling of outliers. Without these, reproducibility cannot be judged.
- Numerous pairwise tests are performed with limited power in subgroups. The Bonferroni correction is mentioned but not consistently applied. Effect sizes and adjusted p-values should be reported, ideally in regression models rather than multiple univariate tests.
- The discussion emphasizes Gln/Glu ratios and diagnostic implications, but these data are not shown. Ratios should be presented in tables/figures, accompanied by statistical testing and ROC analysis for diagnostic performance.
- Authors attribute urea changes to oral microbiome diversity, but no microbiome data were collected. This should be reframed as a hypothesis, not a conclusion.
- Salivary flow rate and protein concentration were not measured. These can alter analyte concentrations. At the very least, this limitation should be acknowledged.
- Revising the title to 'potential salivary indicators' is recommended to avoid overstatement.
Author Response
Responses to reviewer’s comments_2
The authors are grateful to the reviewers for careful consideration of the manuscript and valuable comments. We hope that thanks to the joint work, the manuscript has become better.
- The abstract and methods state BC n=543, FA n=597, HC n=298, with amino acids in a subset (BC n=116, FA n=24, HC n=25). Yet, Table 1 lists BC (n = 660), FA (n = 134), and HC (n = 127). These discrepancies must be resolved, and the sample sizes for each analyte must be clearly stated.
Table 1 contains incorrect data, we have adjusted the corresponding values.
- BC patients are older on average; urea correlates with age and menopause. While stratification by menopause is shown for urea, the amino acid results are not adjusted.
We have added to the text of the manuscript a description of the calculations that allowed us to omit this factor in further discussions: «No correlation with age and menopause status was shown for the concentration of free amino acids in saliva. Thus, for Leu+Ile, the correlation coefficients with age were r=0.0552, r=0.1186 and r=-0.0123 for BC, FA and healthy control, respectively (p˃0.05). The correlation coefficients were for Val r=0.1780, r=-0.0141 and r=-0.0649; for Glu r=0.1767, r=0.0243 and r=0.1615; for Gln r=-0.1034, r=0.1780 and r=0.1464 in the corresponding three subgroups (p˃0.05). Division into subgroups by menopause status also was not show significant differences in the content of free amino acids in saliva, so further calculations were carried out without additional division into subgroups.»
Regression models that adjust for age, menopause, and other covariates (such as renal disease, oral health, diet, smoking, and hydration) are needed. Otherwise, findings may reflect age-related or renal differences rather than cancer.
We collected patient medical history data, so the study included patients without clinically significant systemic pathologies, including kidney disease, diabetes, and others. An oral examination by a dentist was also conducted to rule out the presence of oral diseases or poor oral hygiene, which could affect salivary parameters. The corresponding clarifications are included in the text of the manuscript. We were unable to collect data on smoking, diet, and drinking habits; we have included this information as a limitation of the study.
«Participants were enrolled into the groups simultaneously. Inclusion criteria included female gender, age 30–70 years, the absence of any treatment at the time of the study, including surgery, chemotherapy, or radiation, no signs of active infection (including purulent processes), and oral hygiene. Exclusion criteria included the absence of histological verification of the diagnosis. A physician excluded chronic, inflammatory, and infectious diseases during a routine medical examination. A dental examination was also conducted to rule out any inflammatory oral diseases that could affect the saliva analysis results.»
- Urea and amino acids are influenced by kidney function, diet, and oral conditions. The manuscript lacks data on renal status, oral disease, medications, or dietary intake. These should be reported or at least discussed as limitations.
Kidney and oral diseases were excluded during the experimental design phase, leaving diet, smoking, and fluid intake as uncontrolled variables. We included this information in the study's limitations:
«Study limitations included the lack of information on smoking status, diet and fluid intake, and medication use, which could have influenced saliva composition. We also did not determine the oral microbiota composition in this study, so the conclusions regarding the relationship between salivary urea concentration and the oral microbiome should be considered a hypothesis that requires further verification.»
- The LC-MS/MS and enzymatic assays lack critical QC information: use of isotope-labeled standards, calibration ranges, intra-/inter-assay CVs, replicate analyses, and handling of outliers. Without these, reproducibility cannot be judged.
We have added relevant information to section 2.3.
- Numerous pairwise tests are performed with limited power in subgroups. The Bonferroni correction is mentioned but not consistently applied. Effect sizes and adjusted p-values should be reported, ideally in regression models rather than multiple univariate tests.
All p-values ​​in the manuscript have been adjusted using the Bonferroni correction. We have added a clarification in Section 2.4: «The Bonferroni correction was applied as a p-value adjustment when comparing more than two subgroups: instead of adjusting the alpha level, each p-value was multiplied by the number of tests and the alpha level was left unchanged (p<0.05)».
- The discussion emphasizes Gln/Glu ratios and diagnostic implications, but these data are not shown. Ratios should be presented in tables/figures, accompanied by statistical testing and ROC analysis for diagnostic performance.
We've added information on calculating the Gln/Glu-ratio to the text of the manuscript. The small subgroup sizes preclude calculating diagnostic efficacy, as it's only appropriate to use this ratio for comparing molecular biological subtypes.
«Additionally, the Gln/Glu-ratio was calculated. It was shown that the Gln/Glu-ratio was higher in the saliva of the control group (5.43 [3.30; 10.5]) compared to breast cancer (2.22 [0.84; 5.40], p=0.0094) and fibroadenomas (1.94 [0.89; 6.05], p=0.0184). The Gln/Glu-ratio differed for molecular biological subtypes of breast cancer. Thus, the minimum ratio was characteristic of luminal A (1.26 [0.84; 3.63], p=0.0057), luminal B HER2-negative (2.61 [0.77; 5.27]) and non-luminal breast cancer (3.23 [1.45; 5.40]). For luminal B HER2-positive and TNBC, the Gln/Glu-ratio increased sharply to 8.23 ​​[3.24; 10.9] (p=0.0327) and 11.2 [4.28; 15.2] (p<0.0001) compared with healthy controls».
- Authors attribute urea changes to oral microbiome diversity, but no microbiome data were collected. This should be reframed as a hypothesis, not a conclusion.
We have added this information to the study limitations.
- Salivary flow rate and protein concentration were not measured. These can alter analyte concentrations. At the very least, this limitation should be acknowledged.
We measured both of these indicators and found no significant differences between the subgroups. We have updated the manuscript accordingly in Section 2.3.
«In all subjects, the salivary flow rate (mL/min) was calculated and the concentration of total protein in saliva (g/L) was determined by the reaction with pyrogallol red. No statistically significant differences in these parameters were found between the subgroups; therefore, the data are not presented in the text of the manuscript».
- Revising the title to 'potential salivary indicators' is recommended to avoid overstatement.
We have made a corresponding change to the manuscript title: "Salivary BCAA, glutamate, glutamine and urea as potential indicators of nitrogen metabolism imbalance in breast cancer."

Reviewer 3 Report
Comments and Suggestions for Authors
Sarf and colleagues investigate salivary metabolites as potential non-invasive biomarkers for breast cancer and its molecular subtypes. The topic is relevant and timely, as saliva-based diagnostics are gaining attention for oncology and personalized medicine. The manuscript is generally well structured and presents novel findings. However, major and minor issues need to be addressed before considering publication, as shown below:
A) The introduction could be streamlined to emphasize the rationale for using saliva as a diagnostic fluid in breast cancer.
B) While the study demonstrates interesting associations, the central hypothesis is not clear. The authors should explicitly state whether the work aims to discover diagnostic biomarkers, gain mechanistic insight into nitrogen metabolism, or both.
C) The diagnostic or prognostic utility of the biomarkers is not quantitatively assessed. ROC curve analysis, sensitivity/specificity estimates, or predictive modeling would greatly strengthen the translational impact of the study.
D) Age and menopausal status are acknowledged as confounders, but other factors (e.g., comorbidities, diet, oral health, renal function, and medication use) can also significantly influence salivary metabolites. The lack of adjustment for these variables limits the strength of the conclusions. More details are needed regarding pre-analytical control (e.g., oral hygiene, smoking, dietary restrictions prior to sampling), as these can strongly influence salivary metabolite levels.
E) The discussion links salivary metabolite alterations to oral microbiome changes and tumor metabolism, but these remain speculative. Without direct microbiome or systemic metabolite data, these claims should be toned down or explicitly presented as hypotheses for future research.
F) Figures should be improved for clarity (e.g., axis labels, font size, legends). Some statistical annotations in figures are difficult to interpret.
Comments on the Quality of English LanguageThe manuscript is generally understandable but requires careful English editing to improve readability and flow.
Author Response
Responses to reviewer’s comments_3
The authors are grateful to the reviewers for careful consideration of the manuscript and valuable
comments. We hope that thanks to the joint work, the manuscript has become better.
Sarf and colleagues investigate salivary metabolites as potential non-invasive biomarkers for breast
cancer and its molecular subtypes. The topic is relevant and timely, as saliva-based diagnostics are
gaining attention for oncology and personalized medicine. The manuscript is generally well structured and
presents novel findings. However, major and minor issues need to be addressed before considering
publication, as shown below: 8
9
A) The introduction could be streamlined to emphasize the rationale for using saliva as a diagnostic fluid
in breast cancer.
We have made appropriate changes to the text of the Introduction.
B) While the study demonstrates interesting associations, the central hypothesis is not clear. The authors
should explicitly state whether the work aims to discover diagnostic biomarkers, gain mechanistic insight
into nitrogen metabolism, or both.
We hypothesize that the assessment of urea and amino acid levels in saliva may provide important
information about the characteristics of nitrogen metabolism in breast cancer, including different
phenotypes, which may be useful for both diagnostic and therapeutic purposes.
C) The diagnostic or prognostic utility of the biomarkers is not quantitatively assessed. ROC curve
analysis, sensitivity/specificity estimates, or predictive modeling would greatly strengthen the translational
impact of the study.
In this manuscript, we described the characteristics of nitrogen metabolism using salivary parameters. It
was also shown that both urea and amino acid concentrations depend on the breast cancer phenotype,
making them of limited use for diagnostic purposes. Therefore, the diagnostic evaluation was not shown
to be informative and is not included in the manuscript.
D) Age and menopausal status are acknowledged as confounders, but other factors (e.g., comorbidities,
diet, oral health, renal function, and medication use) can also significantly influence salivary metabolites.
The lack of adjustment for these variables limits the strength of the conclusions. More details are needed
regarding pre-analytical control (e.g., oral hygiene, smoking, dietary restrictions prior to sampling), as
these can strongly influence salivary metabolite levels.
We collected patient medical history data, so the study included patients without clinically significant
systemic pathologies, including kidney disease, diabetes, and others. An oral examination by a dentist
was also conducted to rule out the presence of oral diseases or poor oral hygiene, which could affect
salivary parameters. The corresponding clarifications are included in the text of the manuscript. We were
unable to collect data on smoking, diet, and drinking habits; we have included this information as a
limitation of the study.
E) The discussion links salivary metabolite alterations to oral microbiome changes and tumor metabolism,
but these remain speculative. Without direct microbiome or systemic metabolite data, these claims should
be toned down or explicitly presented as hypotheses for future research.
We included this information in the study's limitations:
« We also did not determine the oral microbiota composition in this study, so the conclusions regarding
the relationship between salivary urea concentration and the oral microbiome should be considered a
hypothesis that requires further verification.»
F) Figures should be improved for clarity (e.g., axis labels, font size, legends). Some statistical
annotations in figures are difficult to interpret.
The corresponding changes have been made to the figures and captions.

Round 2
Reviewer 1 Report
Comments and Suggestions for Authors
The manuscript has been improved, but significant amount of repetition rate is still detected at all the sections.
Author Response
We have revised the manuscript again to reduce the amount of borrowing.
Reviewer 2 Report
Comments and Suggestions for Authors
The authors have satisfactorily addressed the key concerns raised in the initial review. The revised manuscript is substantially improved in clarity and rigor and is now suitable for publication.
Author Response
Thank you for your valuable comments, which helped improve the manuscript.
Reviewer 3 Report
Comments and Suggestions for Authors
The authors answered all the reviewers' questions, and the revised manuscript was improved.
Author Response

(The authors gave the same response as above.)
